

# Effectiveness of virtual reality on activities of daily living in children with cerebral palsy: a systematic review and meta-analysis

YongGu Han[1] and SunWook Park[2]

[1] Department of Physical Therapy, Yonsei University, Wonju, Kangwondo, South Korea
[2] Department of Physical Therapy, Kangwon National University, Samchuk, Kangwondo, South Korea

Corresponding author
SunWook Park,
swpt.park@kangwon.ac.kr

## ABSTRACT

**Background**. No meta-analysis has been conducted on the effect of specific virtual reality (VR) treatment modes on activities of daily living (ADL) in children with cerebral palsy (CP). Therefore, this study aimed to confirm whether VR therapy is effective in improving ADL in children with CP according to subgroups.

**Methodology**. Literature published in the Cumulated Index to Nursing and Allied Health Literature (CINAHL), Embase, the Physiotherapy Evidence Database (PEDro), and PubMed was reviewed, and Risk of Bias 2.0 (RoB 2) was used to evaluate the quality of the literature. A funnel plot was visually observed to confirm publication bias, supplemented with Egger's regression test. Data analysis was performed using R version 4.2.1. Subgroup analysis was performed according to the Gross Motor Function Classification System (GMFCS), the Manual Ability Classification System (MACS), treatment minutes per week, treatment period, age, and RoB.

**Results**. Eleven of 2,978 studies were included, and the overall effect size was 0.37 (95% confidence interval = 0.17–0.57). Regarding GMFCS, effect sizes of 0.41 and 0.33 was observed for the low- and high-function groups, respectively. For MACS, 0.27 and 0.43 were observed for the low and high-function groups. Regarding treatment minutes per week, the values were 0.22, 0.44, and 0.27 in the 1–100, 101–200, and 201–300 min groups, respectively. In the classification according to age, 0.29 was observed for school-age children and 0.98 for preschool children. Lastly, in the classification according to the RoB, 0.52, −0.01, and 0.23 indicated studies with low risk, some concern, and high risk, respectively.

**Conclusions**. The highest effect was observed when VR was applied within 6 weeks of 101-200 per week. Therefore, it is suggested that if the results of this review are applied to children with cerebral palsy in the community, it will be an effective intervention method.

**Systematic review registration**. PROPEROS (registration number CRD42023409801).

# INTRODUCTION

Recent interventions for patients with cerebral palsy (CP) have focused on improving activity, participation, body structure, and function (*Franki et al., 2012*). Notably, the disabilities of children with CP restrict the activities they experience daily, increasing their dependence on caregivers (*Chulliyil et al., 2014*). Thus, improving the activities of daily living (ADL) in children with CP is an important treatment goal to improve children's quality of life in clinical practice (*Franki et al., 2012*; *Chulliyil et al., 2014*).

Various intervention methods have been used in clinical practice to improve ADL in children with CP. Interventions to improve the function of CP are classified based on the international classification of functioning, disability and health (ICF) criteria to improve body function, activity, and participation (*Schiariti et al., 2014*). Manual therapies such as NDT and Bobath are being used to improve the body and structural level, and goal-directed training is being used to improve motor activities and participation in the real world (*Chen, Fanchiang & Howard, 2018*; *Novak et al., 2020*). Whole-body vibration, treadmill, virtual reality (VR) and robotic therapy are devices used clinically to improve the function of CP (*Franki et al., 2012*; *Novak et al., 2020*; *Ghai & Ghai, 2019*; *Han & Yun, 2020*). Regarding interventions for children and junior children, it is essential to maintain attention to treatment by arousing interest (*Bonnechère et al., 2014*; *Chen, Fanchiang & Howard, 2018*). VR is an effective intervention to increase children's motivation. VR provides an environment where a subject feels like the real world and promotes functional activities within the virtual environment (*Parsons et al., 2009*). Specifically, VR can conveniently manipulate the duration, intensity, and frequency and induce task-specific training (*Moreira et al., 2013*). Hence, VR can effectively improve children's functional levels and many studies have reported its positive effects (*Rostami et al., 2012*; *James et al., 2015*; *Atasavun Uysal & Baltaci, 2016*; *Acar et al., 2016*; *Mitchell, Ziviani & Boyd, 2016*; *Tarakci et al., 2016*; *Avcil et al., 2021*; *Choi et al., 2021*; *Jha et al., 2021*; *Wang et al., 2021*).

However, most studies that confirmed the effect of VR on ADL in children with CP were conducted with low-level evidence, and studies that combine conflicting results are lacking. As a result, it is necessary to identify the level of evidence in existing studies and draw broad conclusions using a systematic review and meta-analysis to analyze the impact of the VR intervention on ADL in children with CP (*Borenstein et al., 2021*; *Chandler et al., 2013*). However, most systematic reviews and meta-analyses that confirmed the effects of VR have focused on motor function, body structure, and function, and studies analyzing ADL levels are lacking. For instance, *Chen, Fanchiang & Howard (2018)* studied the effects of VR on upper extremity function, ambulation, and postural control; *Ghai & Ghai (2019)* studied the effects of VR on gait function; and *Abdelhaleem, El Wahab & Elshennawy (2022)* analyzed only the effect on motor coordination. Therefore, it is difficult to define an effective VR intervention for improving ADL because individual studies, including existing randomized controlled trials (RCT), vary in VR application methods and subject characteristics. Additionally, regarding meta-analyses, analysis was not performed based on ADL. Furthermore, most systematic reviews and meta-analyses that confirmed the effect

of VR on children with CP did not confirm the difference in effect size according to the subgroup.

It is necessary to confirm the evidence for the effect of VR on the ADL of children with CP. Therefore, we conducted a systematic review and meta-analysis by classifying studies that confirmed the effects of VR on ADL in children with CP, aiming to propose the best protocol for applying VR to children with CP.

## Survey methodology

This systematic review was performed according to the Preferred Reporting Items for Systematic Reviews and Meta-Analyses flowchart (*Salameh et al., 2020*). Two authors (YG Han, SW Park) independently reviewed the titles and abstracts. After exclusion of duplicates, one author (YG Han) performed a full text review, and the final list of studies was discussed among the writing group (YG Han, SW Park). Conflicts in opinions were settled through discussion and opinions of a physical therapy professor. The study was designed by a researcher well-versed in systematic reviews and meta-analyses and conducted by a professor in the Department of Physical Therapy and a graduate student of physical therapy in the doctoral course.

## Search strategy

This study was designed according to the Participants, Intervention, Comparison, and Outcome guidelines. Participants were children diagnosed with CP. Additionally, the intervention was VR, and conventional physical therapy, no intervention, and other interventions were compared. Moreover, the outcome was the effect of VR on ADL of children with CP. The data search was conducted in CINAHL, Embase, PEDro, and PubMed databases between October 2006 and February 2023, and the language of the literature was restricted to English (Fig. 1).

The MeSH or Emtree term was used as a search strategy, and ((CP) and (VR or game)) were used. The details of the search formula are as follows; ((Reality, Virtual) OR (Virtual Reality, Educational) OR (Educational Virtual Realities) OR (Reality, Educational Virtual) OR (Virtual Realities, Educational) OR (Virtual Realities, Instructional) OR (Instructional Virtual Realities) OR (Instructional Virtual Reality) OR (Realities, Instructional Virtual) OR (Reality, Instructional Virtual) OR (Video game*) OR (Nintendo) OR (Xbox) OR (Xbox kinect) OR (Wii controller) OR (Wii Fit) OR (virtual world)) AND (Cerebral palsy (MeSH Terms)).

## Eligibility criteria

The selection criteria were as follows: (1) studies with children diagnosed with CP; (2) studies comparing VR with other therapy or non-intervention groups; (3) studies in which measurement results were related to ADL; and (4) studies published in English; and (5) RCTs.

The exclusion criteria were as follows: (1) studies with a single experimental design without a control group; (2) non-experimental studies, such as survey studies, case studies, and qualitative studies; (3) non-peer-reviewed gray papers (for example, abstract and

**Identification of studies via databases and registers**

**Identification**

Records identified from: 2948
  Pubmed (n = 2700)
  Pedro (n = 42)
  Embase (n=159)
  CINAHL (n= 47)

**Screening**

Title/abstract reviewed
(n = 197)

Records removed *before screening*:
  Duplicate records removed
  (n = 2751)

Full text reviewed
(n = 55)

Records removed
  Not ADL (n = 33)
  No control group (n= 3)
  No results (n= 4)
  No English (n= 1)
  VR vs VR (n= 1)

**Eligibility**

Reports assessed for eligibility
(n = 13)

Reports excluded:2
  Not RCT (n = 1)
  Inaccurate statistics (n=1)

**Included**

Studies included in review
(n = 11)

**Figure 1** **Search and selection process flowchart.**

poster); (4) studies for which sufficient data were not provided for effect size analysis; and (5) studies with errors in the results presented in tables or figures.

## Data extraction

With the agreement of all the research team members, author names, publication year, Gross Motor Function Classification System (GMFCS), Manual Ability Classification System (MACS), study participants, program type, program effectiveness, and outcomes were recorded for data coding.

## Quality assessment

This study was evaluated using the Cochrane risk-of-bias (RoB 2) tool for randomized trials. Following the Cochrane Handbook Systematic Reviews of Interventions criteria (*Higgins et al., 2019*), two researchers independently performed the quality assessment, discussed it, and reached a consensus. The domains of RoB 2 were as follows: (1) process of randomization, (2) deviations from intended interventions, (3) data of missing outcome, (4) estimation of outcomes, (5) selection of reported results and (6) overall bias.

## Publication bias

A funnel plot was visually observed to confirm publication bias, and the subjective part was supplemented using Egger's regression test (*Borenstein et al., 2021*; *Peters et al., 2006*). In Egger's regression test, the regression line was drawn as an intercept; the closer the intercept was to zero, the smaller the publication bias. However, when the intercept of the regression line was large, and the *p*-value was < 0.1, the publication bias was considered significant (*Peters et al., 2006*). Additionally, Duval and Tweedie's trim and fill method was used for the sensitivity analysis (*Duval, 2005*).

## Data analyses

Data were analyzed using R version 4.2.1 (*R Core Team, 2022*). Additionally, Hedge's g—the corrected effect size—was calculated based on the standardized mean difference in Cohen's d (*Lin & Aloe, 2021*). Correcting Cohen's d value was necessary since it is sensitive to the sample and tends to overestimate the effect size when small (*Lin & Aloe, 2021*). Therefore, Hedge's g value was calculated to obtain the summary value of the effect size, the Z value was calculated to confirm the overall effect size, and the significance level was set at $p < 0.05$ (*Borenstein et al., 2021*). Furthermore, the results were interpreted based on the point estimates; the interpretation criteria for the effect size were assigned as low, medium, and large for point estimates of ≤0.3, approximately 0.5, and ≥0.8, respectively, with a confidence level of 95% (*Han & Yun, 2020*; *Lin & Aloe, 2021*).

## RESULTS

The general characteristics of the included studies are presented in Table 1. Of the 2978 studies evaluated, 11 were finally included, and coding was completed in the order of author, group, number of subjects, type of intervention, type of CP, sex, age, GMFCS score, MACS score, duration, and outcome (*Rostami et al., 2012*; *James et al., 2015*; *Atasavun Uysal & Baltaci, 2016*; *Acar et al., 2016*; *Mitchell, Ziviani & Boyd, 2016*; *Tarakci et al., 2016*; *Avcil et al., 2021*; *Choi et al., 2021*; *Jha et al., 2021*; *Wang et al., 2021*; *Reid & Campbell, 2006*). Additionally, we included 442 participants aged 4.33–11.8 years with spastic and dyskinetic CP types. Furthermore, the VR types included Web-based VR, Nintendo Wii™, Kinect-based virtual reality games, LMC games, and Mitii™. Moreover, ADL evaluation tools, which are dependent variables, included the COPM, PMAL, PEDI, WeeFIM, LIFE-H recreation, and CHAQ according to previous studies and ICF criteria (*Franki et al., 2012*; *Chen, Fanchiang & Howard, 2018*). Before conducting this study, subgroup analyses were decided according to the GMFCS, MACS, weekly treatment minutes, treatment period, and RoB 2, and statistical analyses were performed after data coding was completed. Notably, subgroup variables were based on a previous study (*Wu et al., 2022*). The criteria for dividing GMFCS were defined as high level for stages 1 and 2 and low level for stages 3–5. In the case of GMFCS and MACS, the developmental speed and pattern of stages 1–2 and 3–5 were different (*Ghasia et al., 2008*). In addition, according to previous studies, subgroup analysis was performed based on the length of each session, training session, and age, and differences were found in the effect size (*Lee, Park & Park, 2019*). Finally, in a previous study, classification was performed based on the risk of bias, and significant

differences were also found (*Zhang et al., 2022*; *Ilic et al., 2018*). Therefore, in the present study, subgroups were selected based on various criteria.

Lastly, the age classification criteria were preterm, newborn, infant, toddler, preschool children, school-aged children, and adolescent, as presented by the American Academy of Pediatrics (*Pediatrics, 2020*).

## Assessment of quality

The Risk of Bias (RoB) 2.0 results are shown in Fig. 2. There were five papers with low risk, two with some concern, and four with high risk.

## Publication bias

The funnel plot results were visually observed to evaluate publication bias, and asymmetry was found at low levels. However, Egger's regression test confirmed no publication bias ($t = -1.46$, $p = 0.178$). Additionally, in the sensitivity analysis of trim and fill, four studies were cut and moved from 0.385 ($Q = 10.314$, confidence interval [CI]=0.201–0.565) to 0.484 ($Q = 17.511$, CI=0.483–0.298). Therefore, since the difference was < 0.1, we established that no publication bias in the sensitivity analysis (Fig. 3).

## Homogeneity test for model selection

A homogeneity test was performed to confirm the homogeneity of the included studies, and heterogeneity was observed ($Q = 10.314$, $P = 0.413$, and $I = 3.048$). However, because the distribution of effect sizes in visual observation was inconsistent and the methodological characteristics of the included studies were different, a random effects model was used, and additional analysis was performed through subgroup analysis.

## Combined effect of VR

The summary effect was calculated by deriving the combined effect sizes for each study. The overall effect size of VR on ADL in patients with CP was 0.37 (CI=0.17–0.57), considered a medium effect (Fig. 4), and the difference was statistically significant.

## Effect size according to GMFCS

GMFCS three, four, and five were classified as having a low function, and stages one and two were classified as having a high function. Additionally, studies that did not describe the participants' GMFCS levels were classified as not reported (NR); the results are shown in Fig. 5. Effect sizes of 0.41, 0.33, and 0.27 indicated low-function, high-function, and NR groups, respectively. Lastly, we observed no significant difference in the effect size according to the GMFCS functional level.

## Effect size according to MACS

MACS scores of three, four, and five were classified as low function, and stages one and two were classified as high function. Additionally, studies that did not describe the participants' MACS levels were classified as NR. The results are shown in Fig. 6; effect sizes of 0.27 for low function, 0.43 for high function, and 0.63 for NR. Lastly, the effect size according to the MACS functional level was different.

Han and Park (2023), *PeerJ*, DOI 10.7717/peerj.15964

**Table 1  Study characteristics.**

| Study | Group | Number of participants<br>Intervention | Types of<br>Cerebral palsy | Sex<br>(M/F) | Age<br>Mean (SD) | GMFCS Level<br>MACS Level | Duration | Outcome |
|---|---|---|---|---|---|---|---|---|
| *Reid & Campbell (2006)* | Experimental | 19<br>Web-based VR | Not reported | 12/7 | 9.68 (2.32) | level 1 (7), level 3 (7),<br>level 4 (1), level 5 (4)<br><br>Not reported | 90 min × 8 wk<br>(Day: not reported) | COPM<br>QUEST<br>SPPC |
| | Control | 12<br>Standard care<br>(PT or OT) | Not reported | 8/4 | 9.33 (1.03) | level 1 (4), level 3 (5),<br>level 5(3)<br><br>Not reported | Not reported | |
| *Rostami et al. (2012)* | Experimental | 8<br>VR | Spastic hemiplegia (8) | NR | 7.66(0.96) | Not reported<br><br>Not reported | 90 min 3 d ×4 wk | BOTMP<br>PMAL |
| | Control | 8<br>CIMT | Spastic hemiplegia (8) | NR | 8.33(1.45) | Not reported<br><br>Not reported | 90 min 3 d ×4 wk | |
| *James et al. (2015)* | Experimental | 51<br>Web-based VR | Spastic hemiplegia (51) | 26/25 | 11.8 (2.4) | level 1 (20), level 2 (31)<br><br>level 1 (11), level 2 (39),<br>level 3 (1) | 20–30 min 6 d ×20 wk | AHA<br>AMPS<br>COPM<br>JTTHF<br>MUUL<br>TVPS-3 |
| | Control | 51<br>Standard care | Spastic hemiplegia (50) | 25/25 | 11.10 (2.5) | level 1 (25), level 2 (25)<br><br>level 1 (13), level 2 (37) | 20 wk<br>(Duration: not reported) | |
| *Atasavun Uysal & Baltaci (2016)* | Experimental | 12<br>Nintendo Wii + Traditional PT | Spastic hemiplegia (12) | 8/4 | 9.13 (2.57) | level 1 (9), level 2 (3)<br><br>level 1 (6), level 2 (2),<br>level 3 (4) | 30 min 2 d ×12 wk (VR)<br>45 min 2 d ×12 wk (Traditional PT) | COPM<br>PEDI<br>PBS |
| | Control | 12<br>Traditional PT | Spastic hemiplegia (12) | 2/10 | 10.11 (2.62) | level 1 (10), level 2 (2)<br><br>level 1 (6), level 2 (3),<br>level 3 (3) | 45 min 2 d ×12 wk (Traditional PT) | |
| *Acar et al. (2016)* | Experimental | 15<br>Nintendo Wii (VR) + NDT | Spastic hemiplegia (15) | 8/7 | 9.53 (3.04) | level 1 (6), level 2 (9)<br><br>level 2 (range 1-3) | 15 min 2 d ×6 wk (VR)<br>45 min 2 d ×6 wk (NDT) | ABILHAND-Kids<br>JTHFT<br>QUEST<br>WeeFIM |
| | Control | 15<br>NDT | Spastic hemiplegia (15) | 6/9 | 9.73 (2.86) | level 1 (6), level 2 (9)<br><br>level 2 (range 1-3) | 45 min 2 d ×6 wk (NDT) | |

**Table 1** (*continued*)

| Study | Group | Number of participants / Intervention | Types of Cerebral palsy | Sex (M/F) | Age Mean (SD) | GMFCS Level / MACS Level | Duration | Outcome |
|---|---|---|---|---|---|---|---|---|
| *Mitchell, Ziviani & Boyd (2016)* | Experimental | 51 / Web-based VR | Spastic hemiplegia (51) | 26/25 | 11.3(2.4) | level 1 (21), level 2 (30) | 30 min 6 d ×20 wk | 6MWT Accelerometer Life-H MobQues28 MVPA FSA |
| | | | | | | level 1 (11), level 2 (38), level 3 (2) | | |
| | Control | 50 / Standard care | Spastic hemiplegia (50) | 26/24 | 11.4(2.6) | level 1 (25), level 2 (25) | 20 wk (Duration: not reported) | |
| | | | | | | level 1 (13), level 2 (37) | | |
| *Tarakci et al. (2016)* | Experimental | 15 / Nintendo Wii (VR) + NDT | Spastic hemiplegia (7) Spastic diplegia (5) Dyskinetic (3) | 10/5 | 10.46(2.69) | level 2 (range 1–3) | 20 min 2 d ×12 wk (VR) 30 min 2 d ×12 wk (NDT) | 10MWT 10SCT FRT STST TUG WeeFIM |
| | | | | | | level 1–3 (15) | | |
| | Control | 15 / Balance training (BT) + NDT | Spastic hemiplegia (7) Diplegia (7) Dyskinetic (1) | 9/6 | 10.53(2.79) | level 2 (range 1–2) | 20 min 2 d ×12 wk (BT) 30 min 2 d ×12 wk (NDT) | |
| | | | | | | level 1–3 (15) | | |
| *Avcil et al. (2021)* | Experimental | 15 / Nintendo Wii (VR) + Leap motino controller (LMC) | Spastic hemiplegia (8) Spastic diplegia (4) Dyskinetic (3) | 8/7 | 10.93(4.09) | level 1 (8), level 2 (4), level 3 (1), level 4 (2) | 60 min 3 d ×8 wk (VR) | CHAQ DEI MMDT Dynamometer |
| | | | | | | level 1 (2), level 2 (9), level 3 (4) | | |
| | Control | 15 / NDT-based upper extremity rehabilitation | Spastic hemiplegia (9) Spastic diplegia (2) Dyskinetic (4) | 9/6 | 11.07(3.24) | level 1 (3), level 2 (2), level 3 (6), level 4 (4) | 60 min 3 d ×8 wk (NDT) | |
| | | | | | | level 1 (3), level 2 (8), level 3 (2), level 4 (2) | | |
| *Choi et al. (2021)* | Experimental | 40 / VR for upper-limb + OT | Unilateral (15) Bilateral (25) | 19/21 | 5.33(1.54) | level 1 or level 2 (13) level 3 or level 4 (27) | 30 min 5 d ×4 wk (VR) 30 min 5 d ×4 wk (OT) | 3D motion MA-2 PEDI ULPRS |
| | | | | | | Not reported | | |
| | Control | 38 / OT | Unilateral spastic (19) Bilateral spastic (29) | 19/19 | 4.33(2.31) | level 1 or level 2 (16) level 3 or level 4 (22) | 60 min 5 d ×4 wk (OT) | |
| | | | | | | Not reported | | |
| *Jha et al. (2021)* | Experimental | 19 / VR + PT (Balance) | Bilateral spastic (19) | 14/5 | 8.94(1.92) | level 2 (17), level 3 (2) | 30 min 4 d ×6 wk (VR) 30 min 4 d ×6 wk (PT) | GMFM-88 Kids-mini-Best PBS WeeFIM |
| | | | | | | level 1 (12), level 2 (7) | | |
| | Control | 19 / PT (Balance) | Bilateral spastic (19) | 9/10 | 8.72(1.68) | level 2 (16), level 3 (3) | 60 min 4 d ×6 wk (PT) | |
| | | | | | | level 1 (10), level 2 (9) | | |

Han and Park (2023), *PeerJ*, DOI 10.7717/peerj.15964

**Table 1** (*continued*)

| Study | Group | Number of participants Intervention | Types of Cerebral palsy | Sex (M/F) | Age Mean (SD) | GMFCS Level MACS Level | Duration | Outcome |
|---|---|---|---|---|---|---|---|---|
| *Wang et al. (2021)* | Experimental | 9 Nintendo Wii (VR) + CIMT | Spastic hemiplegia (9) | 3/6 | 8.55(2.09) | Not reported | 135 min 2 d ×4 wk (CIT) 135 min 2 d ×4 wk (VR) | ABILHAND-Kids BOTMP EQ PMAL PSI-SF ToP |
| | | | | | | level 1 (3), level 2 (4) level 3 (2) | | |
| | Control | 9 CIMT | Spastic hemiplegia (9) | 4/5 | 8.57(2.15) | Not reported | 135 min 2 d ×8 wk (CIT) | |
| | | | | | | level 1 (3), level 2 (6) | | |

**Notes.**

GMFCS, Gross motor function classification scale; MACS, Manual ability classification;; VR, Virtual reality; PT, Physical therapy; OT, Occupational therapy; COPM, Canadian occupational performance; QUEST, Quality of upper extremity test; SPPC, Sefl-perception profile for children; CIMT, Constraint-induced movement therapy; BOTMP, Bruiniks-oseretsky test of motor proficiency; PMAL, Pediatric motor activity log; AHA, Assisting hand assessment; AMPS, Assessment of motor and process skills; JTTHF, Jebsen-taylor test of hand function; MUUL, Melbourne assessment of unilateral upper limb function; TVPS-3, Test of visual perceptual skills-3rd edition;; PEDI, Pediatric evaluation of disability inventory; PBS, Pediatric balance scale; NDT, Neurodevelopmental treatment; ABILHAND-Kids, WeeFIM, Functional independence measure for children; 6MWT, 6-minute walk test; LIFE-H, Reaction assessment of life habits recreational domain; MobQues28, Mobility questionnaire 28 item; MVPA, Moderate to vigorous physical activity; NHS, 10MWT, 10-meter walk test; 10SCT, 10-stair climbing test; FRT, Functional reach test; STST, Sit-to-stand test; TUG, Timed up and go test; CHAQ, Childhood health assessment questionnaire; DEI, Duruoz hand index; MMDT, Minnesota manual dexterity test; ULPRS, Upper limb physician's rating scale; EQ, Engagement questionnaire; PSI-SF, Parenting stress index-short form; Top, Test of playfulness.

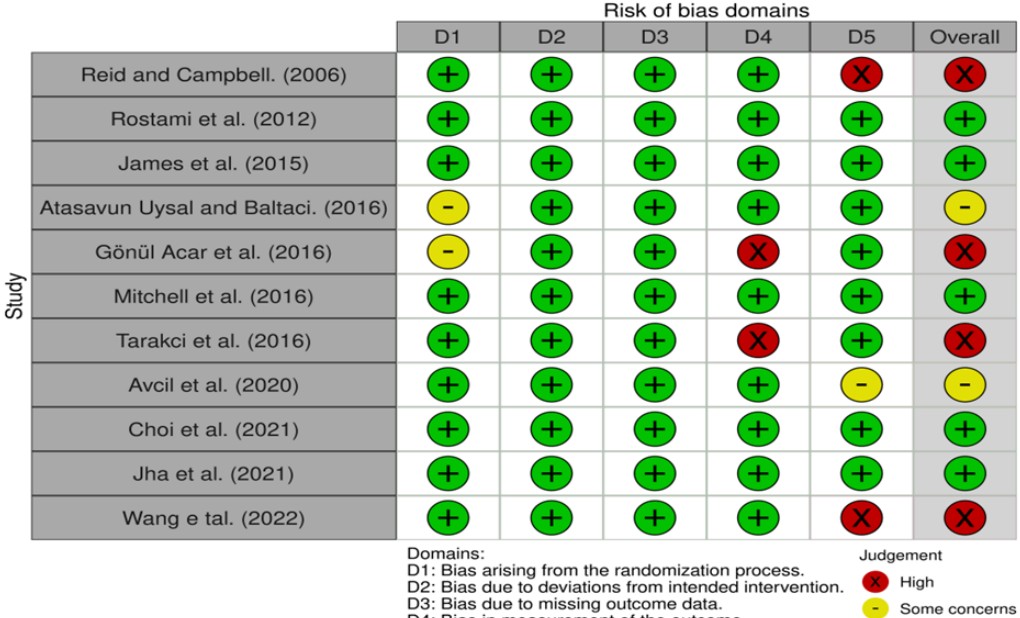

**Figure 2  Results of risk of bias.**

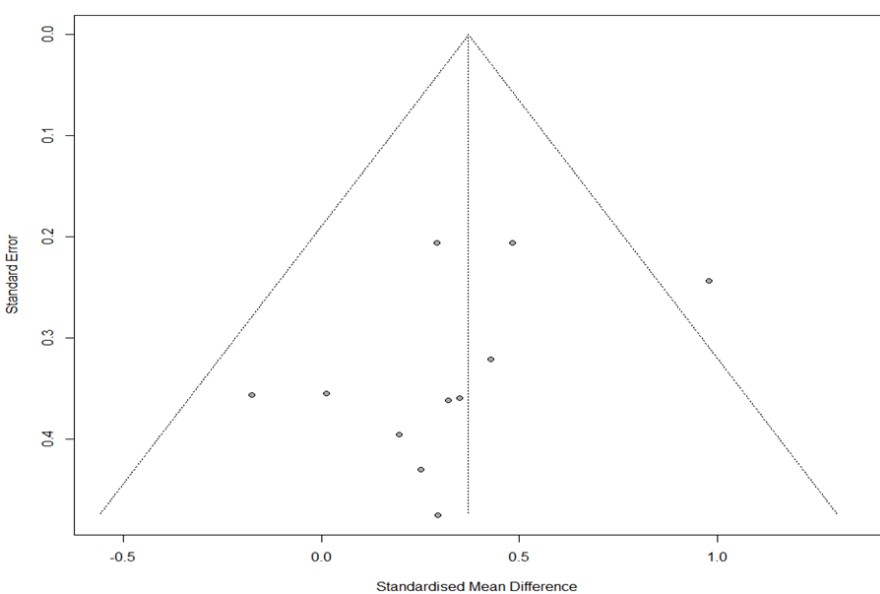

**Figure 3  Funnel plot for publication bias.**

## Effect size according to minutes per week

Treatment minutes per week were classified as 1–100, 101–200, and 201–300 min, with effect sizes of 0.22, 0.44, and 0.27, respectively (Fig. 7). Therefore, a large effect size was observed only in the 101–200 group.

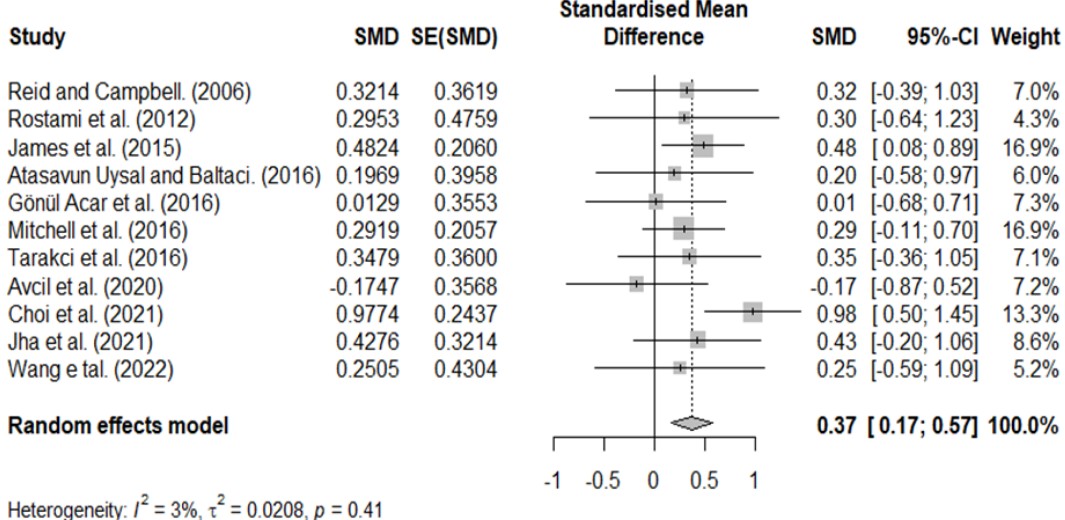

**Figure 4   Homogeneity test.**

**Figure 5   Forest plot according to GMFCS.**

## Effect size according to treatment period

The treatment period was divided into 3–5, 6–8, 9–12, and 20 weeks. A relatively high effect size (0.70) was found at 3–5 weeks, 0.18 at 6–8 weeks, 0.28 at 9–12 weeks, and 0.39 at 20 weeks (Fig. 8).

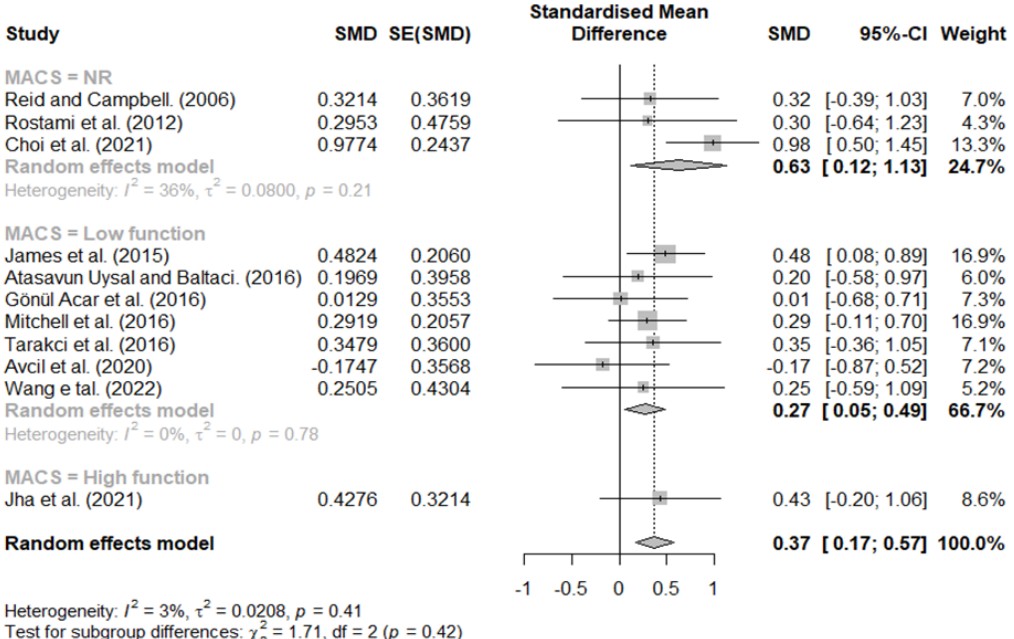

**Figure 6** Forest plot according to MACS.

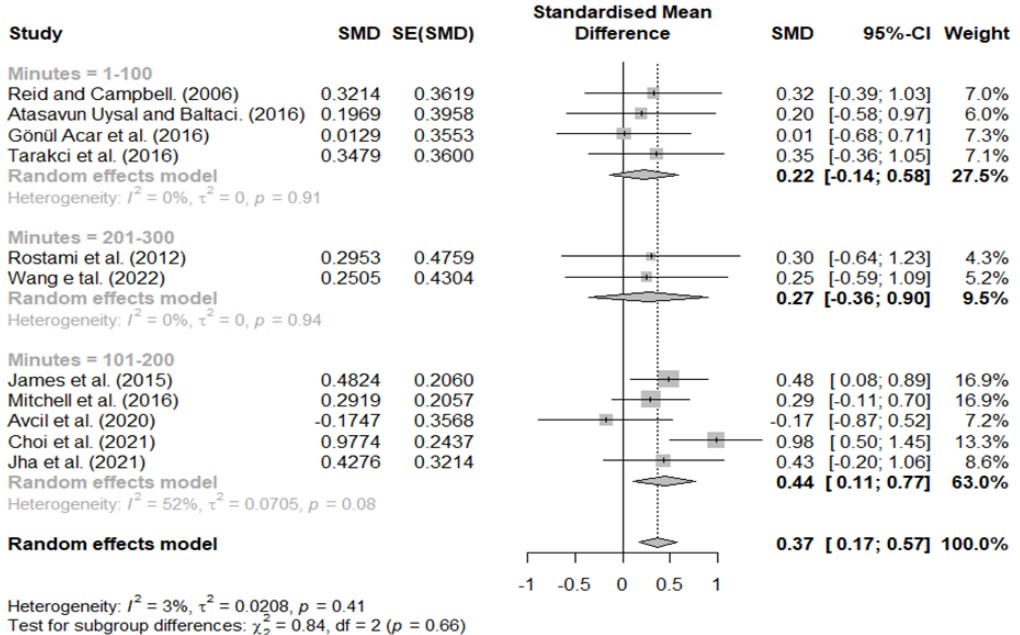

**Figure 7** Forest plot according to minutes per week.

## Effect size according to age

Classification according to age included school-age and preschool children. A significant difference was found with an effect size of 0.29 for school-age children and 0.98 for preschool children (Fig. 9).

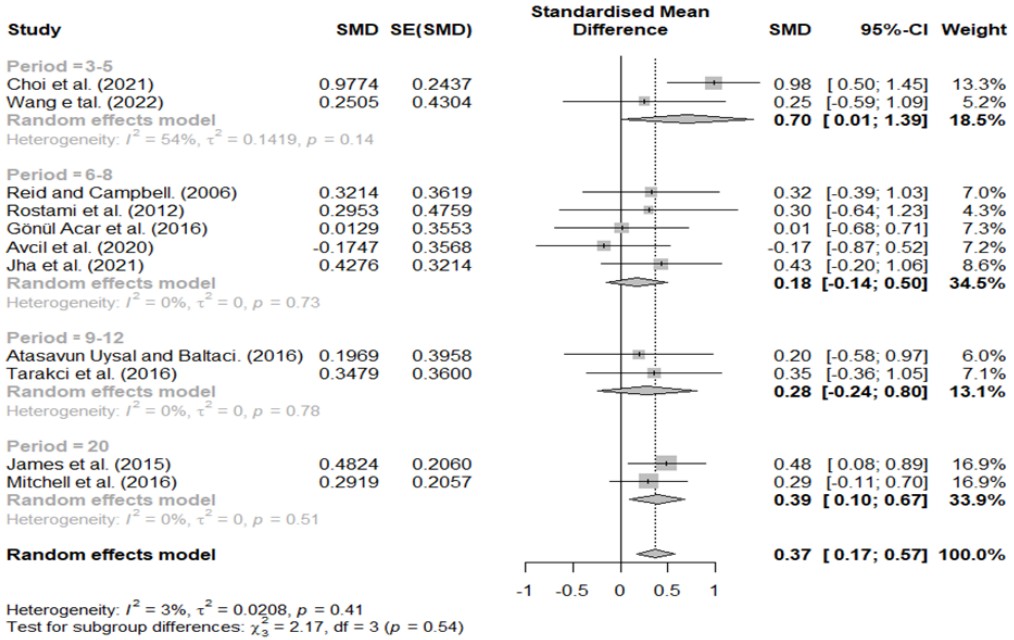

**Figure 8** Forest plot according to treatment period.

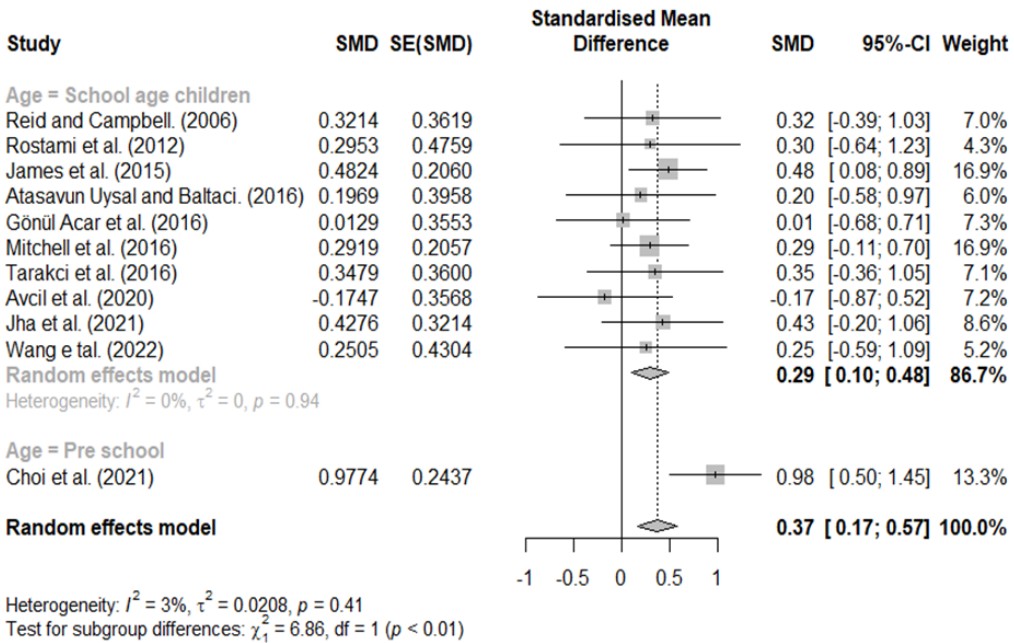

**Figure 9** Forest plot according to age.

## Effect size according to RoB 2

In the classification according to the RoB 2, effect sizes of 0.52, −0.01, and 0.23 indicated studies with low risk, some concern, and high risk, respectively (Fig. 10).

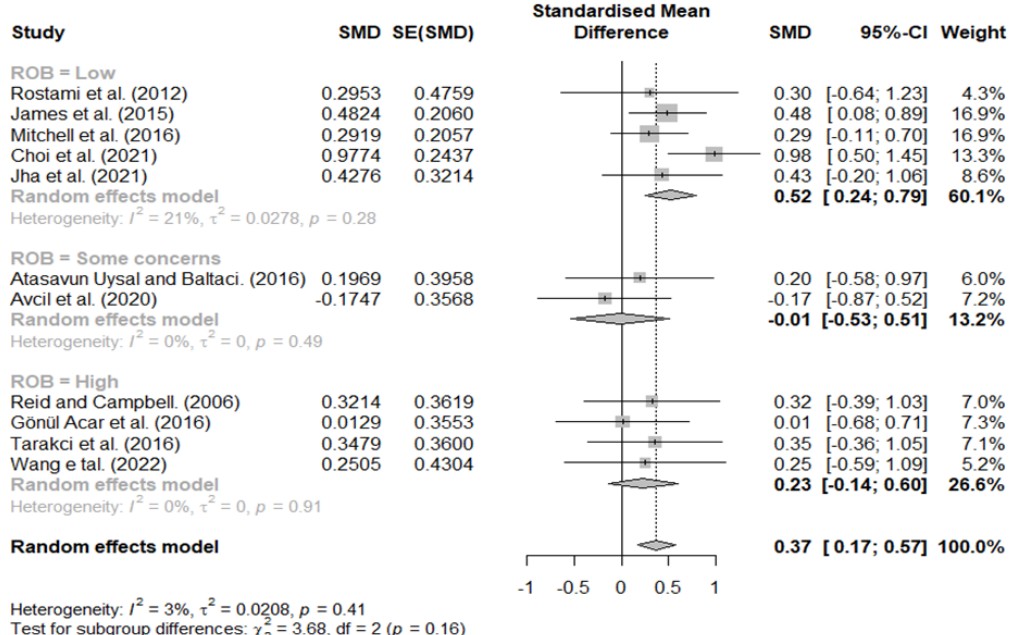

**Figure 10  Forest plot according to ROB.**

# DISCUSSION

This systematic review and meta-analysis were performed to confirm the effect of VR on ADL in children with CP. For the combined effect size of VR, we obtained a medium effect size of 0.37. However, according to previous studies, an effect size of ≥0.5 is considered clinically significant (*Han & Yun, 2020*; *Perera & Cader, 2020*); therefore, our study's results are insufficient to confirm the effect of VR on ADL in children with CP.

However, the effect size differed according to the intervention protocol and participant characteristics, and different results were obtained according to the criteria for dividing the subgroups. First, in the difference according to the GMFCS stage, similar effect sizes were observed in the low- (0.41) and high-function groups (0.33), and the smallest effect size was observed in the NR group (0.27). In regard to MACS, the largest effect size was observed in the NR group (0.63), followed by the high- (0.37) and low-function groups (0.27). Furthermore, for the NR group, although it was difficult to define participant's characteristics, the difference between the high and low groups according to the child's functional level was < 0.1, and there was no significant difference; this may be because VR intervention can select and change the difficulty level according to participant characteristics (*Kuttuva et al., 2006*; *Popescu et al., 2000*). According to previous studies conducted on CP children, studies confirming the effect of virtual reality according to GMFCS and MACS reported that it was effective at all functional levels or there was no difference according to level, and these results are consistent with this review (*Shierk, Lake & Haas, 2016*; *Cherni et al., 2020*). For the reason that these results were derived, most of the subjects included

in this review were in stages 1-3, and it is thought that a ceiling effect occurred (*Johnson et al., 2014*).

In the classification according to treatment minutes per week and the period related to the interval of the VR intervention, a significant difference was found between the groups.

VR interventions allow children to perform tasks similar to real life and promote motivation through games (*Chen, Fanchiang & Howard, 2018*; *Ghai & Ghai, 2019*; *Wu et al., 2022*). Although the VR studies included in this review were adjusted for task difficulty, most intervention programs consistently presented similar games and tasks to children (*Rostami et al., 2012*; *James et al., 2015*; *Atasavun Uysal & Baltaci, 2016*; *Acar et al., 2016*; *Mitchell, Ziviani & Boyd, 2016*; *Tarakci et al., 2016*; *Avcil et al., 2021*; *Choi et al., 2021*; *Jha et al., 2021*; *Wang et al., 2021*). Previous studies reported that it is more effective to apply various tasks to children than provide the same kinds of practice (*Dietze & Kashin, 2019*; *Moreno, 2016*); however, we believe that the studies included in this review had no choice but to control the type of game because of the limitations of the experimental research. Therefore, although the treatment effect was high as children felt a strong interest in the early days, the motivation of those who became accustomed to the game decreased as the period increased; after six weeks, when the children fully adapted to the task, the effect size was halved. However, the duration of exercise increased with an increase in the intervention period; therefore, the effect size increased slightly as it approached 8–20 ''-weeks.

Differences in treatment minutes per week of intervention were 0.22 for 1–100 min, 0.44 for 101–200 min, and 0.27 for 201–300 min, and a medium effect size was observed only in the 101–201 min group. We expected that the effect size would increase as the duration of the intervention increased; however, the effect size decreased in the 201–300 min group. In the previous meta-analysis where VR was applied to CP, the effect size increased as the length of each session, duration of training, and session of training increased (*Zhang et al., 2022*). The reason for the different results from this review in the previous study is that in the previous meta-analysis, not only ADL but also hand grip and gross motor function were included as dependent variables. Unlike hand grip and gross motor function, goal-directed training is required to improve ADL, and task characteristics need to be changed frequently (*Novak et al., 2020*; *Van Merrienboer & Sweller, 2005*). Actually, most of the VR protocols included in this review were conducted in blocked practice rather than random practice (*Rostami et al., 2012*; *James et al., 2015*; *Atasavun Uysal & Baltaci, 2016*; *Acar et al., 2016*; *Mitchell, Ziviani & Boyd, 2016*; *Tarakci et al., 2016*; *Avcil et al., 2021*; *Choi et al., 2021*; *Jha et al., 2021*; *Wang et al., 2021*; *Reid & Campbell, 2006*). In the RCT study, to control the experimental settings, there was no choice but to select a few predetermined games and sequences (*Rostami et al., 2012*; *Avcil et al., 2021*; *Wang et al., 2021*). As a result, the blocked practice was effective for short-term tasks (*Massetti et al., 2018*), but when the duration of the treatment or minute per week was increased, the effect size slightly decreased. So, in future VR research, the types of tasks provided in VR should be configured in various ways based on the perspectives of whole, mental, and random practices (*Van Merrienboer & Sweller, 2005*). In particular, when applying VR to children with CP, studies that consider the practice environment for motor learning in addition to the characteristics of the subject and the duration of intervention are needed (*Novak et al., 2020*; *Jackman et al., 2022*).

We observed a significant difference in the classification according to age (0.29 for school-age and 0.98 for preschool children). This result is consistent with that of *Chen, Fanchiang & Howard (2018)*, who reported an increase in effect size with age in a meta-analysis performed previously in patients with CP. And, this result is consistent with a previous study in which greater effect size was observed in the children aged ≤ 6 years than children aged > 6 years (*Zhang et al., 2022*). Notably, as children with CP age, contracture of muscles and joints becomes more severe, increasing activity restrictions (*Jahnsen et al., 2004*; *Rosenbaum et al., 2002*; *Flett, 2003*). Therefore, the game was thought to be relatively more active during preschool, and a relatively large effect size was observed. However, since the number of studies included in the preschool children group was smaller than that of the school-age children group, it will be necessary to conduct a subgroup analysis, including more studies in future studies.

In the classification according to the quality of the included studies, a medium effect size of 0.52 was observed in the low-risk group, −0.01 in some concerns group, and 0.23 in the high-risk group. Therefore, VR is a sufficiently meaningful intervention in studies with a good design. Notably, even in the same RCT study, there are various defects, such as blindness in a group assignment, selection bias, and performance bias; if a study with severe bias is included, bias will exist in the results (*Higgins et al., 2019*). Therefore, although previous studies have shown a wide variance in effect size, VR is considered a sufficiently meaningful intervention method based on a low-risk group.

Regarding publication bias, asymmetry was found in the visual analysis. However, since the difference in the sensitivity analysis of trim and fill was < 0.1 (0.385−0.484), there was no bias, which was not significant in Egger's regression test. However, besides statistical significance, the Cochrane Handbook and meta-analysis authors do not recommend other statistical testing methods; therefore, care must be taken in the interpretation. Additionally, since the number of studies included in this meta-analysis was small, it is necessary to confirm reproducibility through additional research.

This study had some limitations. First, there are not many high-quality VR studies on CP, so there are limitations in overall effect and subgroup analysis, and it seems necessary to confirm the results through additional meta-analysis in the future. Second, the criteria for the subgroup analysis had a subjective disadvantage; future studies should include more studies and perform a meta-regression using covariates. Third, the classification according to the age of patients with CP was not performed effectively. Therefore, it is difficult to generalize the results to the general CP population because the participants included were primarily school-aged children. However, this study is significant as it is the first meta-analysis to confirm the effect of VR on ADL according to the subgroup when VR was applied to children with CP. Future studies should address the above shortcomings and identify the most effective VR intervention for treating children with CP.

## CONCLUSIONS

This review confirmed the effects of VR interventions on ADL in patients with CP. We observed no difference in the effect of intervention according to GMFCS and MACS for

children. Additionally, the highest effect was observed when VR was applied within 6 weeks of 101–200 min per week. Moreover, when a study with high quality was identified separately, a significant effect of ≥0.5 was observed. Therefore, this study's findings may provide an effective intervention method for patients with CP.

### Funding
The authors received no funding for this work.

### Competing Interests
The authors declare that there are no competing interests.

### Author Contributions
- YongGu Han conceived and designed the experiments, performed the experiments, analyzed the data, prepared figures and/or tables, and approved the final draft.
- SunWook Park conceived and designed the experiments, performed the experiments, analyzed the data, authored or reviewed drafts of the article, and approved the final draft.

### Data Deposition
 The raw measurements are available in the Supplementary File.

### Supplemental Information
Supplemental information for this article can be found online at http://dx.doi.org/10.7717/peerj.15964#supplemental-information.

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
