# Peer review of "Effectiveness of virtual reality on activities of daily living in children with cerebral palsy: a systematic review and meta-analysis"

_PeerJ, doi:10.7717/peerj.15964_

## Round 0.1 · original submission · Minor Revisions

Thank you for your submission to PeerJ. Three reviewers with expertise in this area have now given feedback on your manuscript and highlighted several minor edits that would strengthen the manuscript. I look forward to reading a resubmission of your article in due course.

·

Basic reporting

no comment

Experimental design

I think the authors need to explain
Before conducting this study subgroup analyses were performed according to the GMFCS, MACS, weekly treatment minutes treatment period, and Rob-2

It seems not logical to create all these subclasses. Please introduce why these are relevant and needed.

Validity of the findings

One important factor is the time spent on training and improving motor skills/activities.
Based on the amount of time for training the authors should discuss the number of minutes per week in very detail. In the discussion it is not presented and discussed

I would like to see more in-depth discussion on all parameters and compare these with other motor learning studies outside the field of VR

I think the authors can answer these questions and would like to see the result back

best

Additional comments

no comment

·

Basic reporting

Q1: Line 47, you mention that "Various intervention methods have been used in clinical practice to improve ADL in children with CP", please add what methods are available? How do these methods relate to the methods involved in this study? Please explain.
Q2: In the Survey methodology section, how was the dispute resolved in the literature search? Please add.
Q3: Line220, you mention that “according to previous studies, an effect size of g0.5 is considered clinically significant”, please add the source of the study?
Q4: Line274, you mention that “the number of included literature was insufficient”, why? How to explain this limitation? Is this flaw to negate your research process?
Q5: The language in this paper is noteworthy for its extensive use of the pronoun "therefore"(Line45, 53,57,64,71).

Experimental design

Accuracy

Validity of the findings

Accuracy

·

Basic reporting

The aims and methods of the study are described thoroughly. It was easy to understand what the authors did, how they did it, and what they found.

Everything is as clear as it can be.

Experimental design

There is one issue that the authors didn't mention the year range of their database results. I thought they searched only 2023 papers but the tables told me the opposite. It would be beneficial to write this detail to the manuscript.

Validity of the findings

Written in detail, and left no questions for the reader. I have nothing to add

Additional comments

This study is a systematic review of VR interventions' effects on the ADL skills of children with CP. As a researcher in the VR area, I read a lot of articles, and I think this study was written extremely well.

---

## Round 0.2 · accepted · Accept

I am pleased to inform you that the remaining reviewers have agreed that your manuscript is worthy of publication in PeerJ. Congratulations!

·

Basic reporting

acceptable structure and adequate adaptations.

Experimental design

well performed

Validity of the findings

meets the standard of the journal

Additional comments

acceptable for publication

·

Basic reporting

The text has been revised and embellished to use clear and unambiguous professional English. The supplementary literature references provided demonstrate the context of the study adequately. The structure of the article, figures and tables are largely standardised, and relevant data is shown in the supplementary material. I consider the manuscript acceptable for publication.

Experimental design

The experimental design was strictly adhered to the standards of systematic reviews and meta-analyses.

Validity of the findings

The manuscript provides an excellent overview of a systematic exposition of the issues in the field. The paper provides all the underlying data; these are robust, statistically sound and controlled. The conclusions of the study are clearly stated and closely related to the original research question.

Additional comments

I recommend that the editor-in-chief accept and publish this manuscript.